# A Three-Way Interaction of Sex, PER2 *rs56013859* Polymorphism, and Family Maltreatment in Depressive Symptoms in Adolescents

**DOI:** 10.3390/genes14091723

**Published:** 2023-08-29

**Authors:** Catalina Torres Soler, Sofia H. Kanders, Mattias Rehn, Susanne Olofsdotter, Cecilia Åslund, Kent W. Nilsson

**Affiliations:** 1Centre for Clinical Research, Region Västmanland, Uppsala University, 721 89 Västerås, Sweden; 2Department of Psychology, Uppsala University, 751 05 Uppsala, Sweden; 3Department of Public Health and Caring Sciences, Uppsala University, 751 05 Uppsala, Sweden; 4Department of Neuroscience, Uppsala University, 751 05 Uppsala, Sweden; 5The School of Health, Care and Social Welfare, Mälardalen University, 721 23 Västerås, Sweden

**Keywords:** adolescents, depression, sleep, suicidal ideation, child abuse, association study, genetic

## Abstract

The prevalence of depressive symptoms in adolescents is 12–18% and is twice as frequent in females. Sleep problems and thoughts of death are depressive symptoms or co-occurrent phenomena. Family maltreatment is a risk factor for later depressive symptoms and the period circadian regulator (PER) has been studied in relation to neurotransmitters, adaptation to stress, and winter depression. The purpose of this work was to study the relation of the three-way interactions of sex, PER2 *rs56013859*, and family maltreatment in relation to core depressive symptoms, sleep complaints, and thoughts of death and suicide in self-reports from a cohort of Swedish adolescents in 2012, 2015, and 2018. Cross-sectional and longitudinal analyses with linear and logistic regressions were used to study the relationships to the three outcomes. The three-way interaction was related to core depressive symptoms at both baseline and six years later. In contrast, the model did not show any relation to the other dependent variables. At 13–15 years, a sex-related differential expression was observed: females with the minor allele C:C/C:T exposed to family maltreatment showed higher levels of core depressive symptoms. Six years later, the trend was inverted among carriers of minor alleles.

## 1. Introduction

### 1.1. Depression

In 2019, the estimated worldwide prevalence of years lived with disability (YLDs) caused by depression in the population aged 15 to 49 years was 3.6% [1]. A YLD is equivalent to one year lost in life due to disability or disease. According to the Diagnostic and Statistical Manual of Mental Disorders (DSM 5), major depressive disorder (MDD) is characterized by the presence of five or more depressive symptoms during at least 2 weeks, with at least one being depressed mood or loss of interest or pleasure present at the time of assessment. The basic symptoms are depressed mood and/or loss of interest or pleasure, most of the day on an almost daily basis; a significant change in weight (more than 5%) or decreased or increased appetite nearly every day; sleep disturbances; psychomotor symptoms, such as decelerated thoughts and reduction of physical activity or agitation, energy reduction or loss of energy almost daily; feelings of worthlessness or guilt; decreased ability to concentrate or to think; thoughts of death and/or thoughts of suicide. For diagnosis, these symptoms must cause a loss of function in various areas [2] and should not be due to other diseases or substance use [2].

Fried et al. [3] in their study about the depressive criteria, indicated that the DSM symptoms mentioned above, and other symptoms such as anxiety/tenseness, mood reactivity, diurnal variation, loss of sexual interest, panic/phobia, gastrointestinal problems, somatic complaints, sympathetic arousal, interpersonal sensitivity, and paralysis, do not differ on centrality in the measurement of their relationship with all symptoms. Some additional research shows significant fluctuations over time for a majority of the nine criterion symptoms of major depression according to DSM-5, including sleep disturbances [4]. These fluctuations are suggested to be related to variations in the severity of depression rather than changes in individual symptom profiles [4]. The prevalence of depressive symptoms in adolescents is 11.7–18%, with symptoms being more frequent in females (25%) than males (10%) from the age of 13 years [5,6,7]. Depressive symptoms change over time, indicating a continuum, and the relation to stressful events diminishes relative to the number of episodes [8]. These sex differences in the prevalence of depressive symptoms and multiple reasons can be attributed to the differential course of the symptoms, help-seeking, risk of experiencing stressful events, and vulnerability [7].

### 1.2. Thoughts of Death and Suicide in Depression

Depressive symptoms have been correlated with suicidal ideation and suicide attempts [9,10]. Depression is also considered a risk factor for a severe suicide attempt [11]. Moreover, in adolescents, irritability has been related to suicidal behavior [12]. Suicidal ideation (SI) is a suicide risk factor that can be modified [13]. Nonetheless, SI is considered a predictor of lifetime risk of suicide [14,15]. During adolescence, reports of suicidal thoughts and behaviors increase compared with adults, indicating it is a critical developmental period for suicidal thoughts [14,16,17]. In 2014, Miranda et al. [14] found a prevalence of 11% of self-reported SI during the past 3 months in a New York City high school sample of 506 participants aged 12–21 years. In 2008, Nock et al. [16] found the SI prevalence to range from 3.1 to 15.9% across the world. Among the individuals with SI, the probability of making a suicide plan was 33.6% and the probability of ever carrying out a suicide attempt was 29% [16].

Other risk factors for SI are being female [18,19,20], adolescence, and/or young adult [17], and suffering from a mood disorder [16]. Environmental risk factors are childhood maltreatment, such as sexual, physical, and emotional abuse [21,22,23,24], family environment and integrity, and low socioeconomic status [18]. Sleep disturbances are another risk factor for SI and suicidal behaviors [25,26,27,28]. Females who are exposed to adverse childhood experiences show more internalizing symptoms, including SI, sleep difficulties, and depression, than males, who show more externalizing symptoms, such as alcohol consumption [29].

### 1.3. Sleep Complaints in Depression

As one of the diagnostic criteria for major depression is sleep problems [30], it is important to understand the value of sleep for this common disorder. It is common for individuals with depression to experience sleep problems [31,32,33].

Sleep symptoms have a weak connection with other depressive symptoms, indicating a separate construct for the sleep problems [3]. Previous research shows that circadian rhythm alterations are correlated with mood disorders [34]. About one-third of the population report sleep problems to some degree [35].

Sleep is an essential factor for children and adolescent behavioral, emotional, physical, and intellectual development, as well as being important for learning and memory [36]. Reaching adolescence is associated with changes in sleep behavior and sleep architecture [37,38], and sleep problems are common for children and adolescents [39]. A bidirectional relationship between sleep and health is common [37].

Some research has suggested that the relation between sleep problems and suicide might be mediated by depressive symptoms [40], or that sleep problems may be a critical factor leading to depressive symptoms and suicidal thoughts [41].

### 1.4. Family Maltreatment

Child maltreatment is defined by the World Health Organization as “the abuse and neglect that occurs to children under 18 years of age”, and includes physical, emotional, and/or sexual abuse, and rejection [42]. It is a complex global issue with long-term consequences and the prevalence data is unknown in many countries [42]. The global estimation of maltreatment prevalence is 11.8% [43]. In a survey of 4503 children and youth, maltreatment was present in 12.1% of the sample, and characterized by emotional abuse (5.6%), physical abuse (4%), neglect (4.7%), and sexual abuse (0.1%). Of those affected, 23% were exposed to two or more types of mistreatment [44]. Maltreatment has long-term negative effects in the development of neurological systems, impulse control, emotional regulation, attention, and relationships [45], and has been identified as risk factor in the development of mood disorders [46]. Possible explanations are the subsequent neuroendocrine dysregulation changes and the interactions with sensitive gene variants [47,48].

Additionally, there has been a positive association between adverse experiences and the development of depressive symptoms in some ethnic groups, such as Ghanaians and Moroccans [49]. Although females report higher rates of maltreatment [49], the evidence to identify sex differences in the effects of maltreatment is not yet considered sufficient [50]. Another aspect to be considered is the effect of family interaction patterns and family composition [7,51]. Additional possible stressors in the family in relation to adolescent depression include conflict, emotional closeness, and family mobility [52].

### 1.5. Circadian Rhythms and PER2

The circadian rhythm of mammals is regulated through a complex negative feedback loop that includes transcription of the period circadian regulator 2 (PER2) gene [53,54]. Through interactions with nuclear receptors, PER2 can affect the circadian rhythm in addition to glucose and fatty acid metabolism, temperature, inflammation, and the dopaminergic system [55]. The latter effect is suggested through PER2 regulation of monoamine oxidase A (MAOA) activity leading to increased dopamine levels [56]. PER2 is expressed in a circadian pattern in regions such as the ventral tegmental area and the arcuate nucleus [55].

*Rs56013859* is a single nucleotide polymorphism of the gene PER2. It is also called Spanagel–Albrecht or SNP #10870 [57,58,59]. It is located on chromosome 2 with reference chr2:238276865 (GRCh38.p14) [60]. This is an intron variation with alleles (C:C), (C:T), or (T:T) [61]. The global frequency of allele C is about 0.011 to 0.12 [60]. The importance of the study of introns lies in their potential direct and indirect effects [62,63].

Mutations in PER2 have been linked to a disorder associated with very early sleep onset and offset called familial advanced sleep phase syndrome 1 [64,65]. However, single gene disorders are an exception; most are complex and depend on a combination of genetic and environmental factors as well as their interactions, and evidence linking circadian gene variants directly to timing of sleep and wakefulness is scarce [35]. Moreover, PER2 has been associated with sleep in adaptation to stress [66], and with depression vulnerability [54], in particular winter depression [67]. In animal models, Hampp et al. (2008) found less immobility in the swimming test, which screens levels of depression, in mutant PER2^Brdm1^ mice than in wild-type mice [56]. This behavior is possibly regulated by dopamine activity [68]. However, when mice are exposed to chronic stress, they show less PER2 expression and more depressive-like behavior [69,70]. Melhuish et al. [32] did not find a relationship between PER2 and MDD in humans, but rather than entirely discard any involvement from this gene in MDD, they indicated that further studies are needed [32].

*Rs56013859* polymorphism has been studied in relation to addictions [71]. When investigating the relationship to stressful life events and alcohol use in young adults, Blomeyer et al. [72] found that subjects with the minor allele G were less prone to alcohol use than homozygotes for the major allele A. Moreover, the *rs56013859* A:A genotype has been associated with sleep problems and increased alcohol use among adolescent males [58].

To our knowledge, the *rs56013859* polymorphism has not yet been studied in relation to depressive symptoms in adolescents. However, due to the relationship between alcohol consumption and depressive symptoms in adolescents [73], we considered studying the relationship of the three-way interaction with depressive symptoms.

### 1.6. Aim

The aim of this study was to analyze the three-way interaction of sex, *rs56013859* polymorphism, and family maltreatment with depressive symptoms in adolescents from the Survey of Adolescent Life in the Västmanland Cohort (SALVe cohort).

The study questions were:
Is there an interaction between *rs56013859* polymorphism and sex in combination with family maltreatment in adolescents as predictor of: (a) core depressive symptoms, (b) sleep complaints, and (c) thoughts of death and suicide?If an interaction effect is present, for whom and at which level of family maltreatment is it present?Does the effect of this interaction remain over time?

## 2. Materials and Methods

### 2.1. Sample

The SALVe cohort [74] included 5233 adolescents born in 1997 and 1999, living in Västmanland in 2012. Of these, 533 were excluded because they had lived in Sweden <5 years, had mental disabilities, severe illness, or language difficulties, or did not have legal guardian consent and were younger than 15 years old (Figure 1). The adolescents were contacted by regular mail with an invitation to take part in the study and a self-reported questionnaire. Those who had difficulties that prevented them from answering the forms were excluded (*n* = 5 with mental disability and *n* = 138 with language difficulties).

In total, 1834 participants signed an informed consent form at the first assessment in 2012, provided consent from their legal guardian if they were <15 years at the start of the study, and completed the questionnaire.

In 2015, 1834 participants from 2012, together with 500 non-responders from 2012 were invited to participate, giving a total of 2334. Of these, 690 did not respond and 1644 participated, and of them 1575, 85% were followed up from 2012.

In 2018, 1118 of the 2015 participants, and 276 adolescents who were included in 2012 but not in 2015, were invited to participate. Of these, 708 did not respond, giving a total of 1212 participants.

The study was conducted in alignment with the Declaration of Helsinki and approved by the Ethical Review Board of Uppsala (dnr 2012/187). A description of the study sample is presented in Table 1.

### 2.2. Procedures

#### 2.2.1. Assessment of Depressive Symptoms

At all three time points, the participants self-rated experiences of depressive symptoms during the past two weeks (Yes or No) using the Depression Self-Rating Scale (DSRS) [75]. We also added the irritability criterion based on the adolescent sample [2,76,77,78]. The DSRS consisted of 14 questions based on the DSM-IV A criteria for MDD. In our study, only eight questions on the core symptoms of depression based on the DSRS [75] and criteria proposed by Hieronymus et al. [79] and Trivedi et al. [80] were selected, to enable separate analysis of the effect of the interaction on sleep complaints. The selected questions were:

Over the last two weeks…

Have you felt down, sad, or empty, almost all the time, almost every day?Have you felt annoyed, angry, or upset almost all the time, almost every day?Almost every day, have you felt disinterested in most things or found it difficult to enjoy the things you normally enjoy?Have you been feeling weak, tired, or low on energy?Has your self-confidence been worse than usual?Have you felt guilty or worthless?Have you had difficulty thinking or concentrating?Have you had thoughts about death, or have you thought that it would be better to be dead?

Self-rated answers (No = 0, Yes = 1) were added together to give a core depressive symptoms index ranging from 0 to 8. Questions about weight or appetite, sleep, and psychomotor agitation or retardation were excluded.

Cronbach’s α for the core depressive symptoms index was 0.815 in 2012, 0.835 in 2015, and 0.860 in 2018. In a principal component analysis (PCA), where factors were extracted based on eigenvalue 1, all three time points showed one component, therefore no rotation took place. Eigenvalues and explained variance were 3.57/44.6%, 3.78/47.2%, and 4.06/50.8% in 2012, 2015, and 2018, respectively.

Two questions were used to assess thoughts of death and suicide in 2012 and 2018. The first question was the DSRS item about suicidal thoughts included in the core depressive symptoms index, and the second was “…have you had recurring thoughts of taking your own life?” Both had response options of No = 0 or Yes = 1. A new variable, “Thoughts of death and suicide” was coded as Yes = 1 if the answers to both questions were yes, and otherwise coded as No = 0.

#### 2.2.2. Assessment of Sleep

At all three time points, experiences of sleep-related complaints during the past three months were collected. The 18 items in 2012, 19 items in 2015, and 16 items in 2018 were based on the Karolinska Sleep Questionnaire (KSQ) [81]. The six possible responses ranged from “never” (0p) to “almost always/5 times per week” (5p). To evaluate the consistency of the factor structure of the KSQ over time among this adolescent population, we selected the items available at all three time points, as follows.

Have you been bothered by the following complaints during the past three months:Difficulties falling asleep;Difficulties waking up;Repeated awakenings with difficulties falling asleep again;Not well-rested on awakening;Premature (final) awakenings;Disturbed/restless sleep;Feelings of being exhausted at awakening;Sleepy during school/work;Sleepy during leisure time;Involuntary sleep episodes during school/work;Involuntary sleep episodes during leisure time;Need to fight off sleep to stay awake.

These were included in a PCA where factors were extracted based on eigenvalue 1 using varimax rotation. The PCA revealed two factors in 2012 with an initial eigenvalue of 5.1 explaining 42.4% of the variance for factor 1 and 1.3/10.8% for factor 2. In 2015, three components (5.5/46.0%, 1.5/12.1%, 1.1/9.1%) were revealed. In 2018, three components (5.4/44.6%, 1.5/12.8%, 1.1/9.4%) were revealed. Based on the discrepancy between the factor analyses and the proposed indexes, all items were instead included in a sleep complaints index, one for each time point. A reliability test of the 12 items revealed a Cronbach’s α of 0.89 in 2012, 0.87 in 2015, and 0.88 in 2018.

#### 2.2.3. Assessment of Family Maltreatment

Family maltreatment was based on the adolescents’ answers (in 2012) to the four following questions about violence in the family at any time [82]: 1. Have there been difficult and upsetting arguments between your parents?; 2. Has it happened that one of your parents pushed, hit, or used other violence against the other?; 3. Have you ever been mentally abused (e.g., mocked, insulted) by one of your parents?; 4. Has it happened that one of your parents pushed, hit, or used other violence against you?

Each question had six possible responses ranging between 0 to 5 as follows: No or has not occurred = 0; Yes, less than once a year = 1; Yes, once a year = 2; Yes, once a month = 3; Yes, once a week = 4; and Yes, every or almost every day = 5. The total sum of the answers was calculated into the family maltreatment index (0–20). Cronbach’s α was 0.57 and the PCA showed one component, with an eigenvalue 2.1 and 51.2% of explained variance in 2012.

#### 2.2.4. Genotyping

DNA samples were collected in 2012 through a self-collection kit (Oragene^®^ DNA, DNA Genotek, Ottawa, Ontario, Canada). DNA extraction was performed from 200 μL of saliva using Kleargene^TM^ (LGC, Biosearch Technologies, UK). Of the eligible study candidates, 86.5% were genotyped. Rs56013859 C < T polymorphism was assessed using KASP™ (LGC, Biosearch Technologies, UK), hereafter called *rs56013859* C and T.

Due to the low frequency of the homozygote polymorphism C:C in the population in 2012 (n = 25, 1.4%), this was added to C:T (n = 364, 19.8%) and the variable was dichotomized as C:C/C:T and T:T.

#### 2.2.5. Sex

Sex was registered according to personal identity number, which reveals the individual’s biological sex. The reference group was male.

#### 2.2.6. Age

Age was calculated based on each participant’s personal identity number and used for descriptive statistics.

### 2.3. Statistical Analyses

Chi square analyses [83] were conducted on categorical variables and the Mann–Whitney U test [83] was used for continuous variables to assess differences in age, genotype, family maltreatment index, core depressive symptoms index, sleep complaints index, and thoughts of death or suicide between groups separately by sex.

Cronbach’s α [83] and PCA [83] were used to study the consistency and composition of the scales.

#### 2.3.1. Cross-Sectional Analyses

Linear regressions with a robust estimator [84] were performed to study the three-way interactions of sex, *rs56013859*, and family maltreatment measured in 2012 in relation to the core depressive symptoms index and sleep complaints index in 2012 [85].

Additionally, binary logistic regression [83] was used to study the interaction model with sex, *rs56013859*, and family maltreatment in relation to thoughts of death and suicide in 2012.

For all models, the reference groups were male and genotype T:T.

Significant interaction effects were graphically represented, comparing females with males and genotypes.

In addition to the levels of significance of the interaction, a moderated moderation model 3 with heteroskedasticity-consistent inference with a robust estimator [84] was performed using PROCESS [86,87] (Figure 2).

#### 2.3.2. Longitudinal Analyses

To determine whether the effect of the interaction remained over time, additional separate analyses were performed using the core depression symptoms index (2015 and 2018), sleep complaints index (2015 and 2018), and thoughts of death and suicide (2018) as outcomes in the same manner as in the cross-sectional analyses.

To ensure the quality of the results, additional analyses were carried out using analysis of variance with heteroskedasticity-consistent variance estimator (Davidson–MacKinnon) HC3 [88] as an alternative method in cross-sectional and longitudinal analyses [83].

Since the study was explorative, a correction for the use of multiple tests (i.e., Bonferroni) was not used [89].

An estimation of the statistical power for gene–environment interactions was calculated, using Quanto [90]. The minor allele frequency for *rs56013859* was 0.12. The power to detect a marginal coefficient of determination (R^2^) of 1%, typically seen in observational association studies [91], was 98% for the interaction of *rs56013859* and family maltreatment, assuming a significance level of 0.05 and a two-sided test.

Complementary analyses to control the significance of the interaction were conducted in the proposed model of the three-way interaction substituting PER2 rs56013859 with AUTS2 *rs69343555* without finding statistically significant results [92].

The software used was the Statistical Package for the Social Sciences SPSS (IBM SPSS Statistic for Windows, version 29, IBM Corp., Armonk, NY, USA) and PROCESS v4.2 by Andrew F. Hayes. A *p* value < 0.05 was considered statistically significant [93].

## 3. Results

### 3.1. Crude Analyses

Table 1 shows that at all three time points of measurement, females were represented in higher proportions than males. No differences in age between the sexes were found. Females were more likely than males to report core depressive symptoms. Females presented a higher frequency of thoughts of death or suicide than males in 2012, whereas no such difference between sexes was found in 2018. Females reported significantly higher sleep complaints at all three time points. No differences between the sexes were found for family maltreatment and *rs56013859*.

### 3.2. Cross-Sectional Analyses

#### 3.2.1. Linear Regression with Outcome Core Depressive Symptoms

The overall regression model of the three-way interaction of sex, *rs56013859*, and family maltreatment in relation to core depressive symptoms in 2012 was significant (χ^2^(7, 1634) = 231,685, *p* < 0.001). Importantly, the three-way interaction term was significant, whereas the direct effect of *rs56013859* was not (see Table 2).

The results indicate that female carriers of *rs56013859* C:C or C:T that were exposed to family maltreatment showed more depressive symptoms than females with the T:T genotype and males irrespective of genotype. Moreover, a differential sex-specific effect was observed in relation to core depressive symptoms with a smaller increase trend in males, being lower in carriers of C:C or C:T (see Figure 3).

#### 3.2.2. Moderation Model 3 with Outcome Core Depressive Symptoms Index in 2012

A moderation model 3 [87] with outcome variable Y = core depressive symptoms index in 2012 and predictors as described in Figure 2 was completed (references; sex, male and *rs56013859,* T:T). With n = 1634, *R*^2^ = 0.132, *p* < 0.001, the addition of the interaction showed a significant change to *R*^2^ = 0.003, *p* = 0.032. In this model, the main effect of sex and family maltreatment was statistically significant with *p* < 0.001, and the interaction of *rs56013859,* sex, and family maltreatment was significant with *t =* 2.146 (95% CI 0.032 to 0.713), *p* = 0.032.

The Johnson–Neyman method was used to identify the level of family maltreatment at which the conditional effect of sex by *rs56013859* on depressive symptoms transitioned from non-significant to significant. Results showed this level to be 1.110, with 19.28% of the sample reporting family maltreatment at or above this level.

#### 3.2.3. Linear Regression with Outcome Sleep Complaints Index, and Logistic Regression with Outcome Thoughts of Death and Suicide in 2012

Analyses did not show statistical significance of the interaction effect and can be seen in the Appendix A, Table A1 and Table A2.

### 3.3. Longitudinal Analyses

#### 3.3.1. Outcomes in 2015

Linear regression analyses with outcomes of core depressive symptoms index and sleep complaints index did not show statistical significance of the interaction effect and can be seen in Appendix A, Table A3 and Table A4.

#### 3.3.2. Outcomes in 2018

A significant relationship between the studied interaction and the core depressive symptoms index was found, as can be seen in Table 3.

In this regression model, only sex had a direct relationship to core depressive symptoms. However, as in 2012, the three-way interaction relation to the core depressive symptoms index was significant. Noteworthy is that the group of males carrying C:C or C:T showed more elevated scores on the core depressive symptoms index when they were exposed to family maltreatment, a reversed trend to that observed in 2012, while females carrying the C:C or C:T presented a lower index.

The results of the regression are also shown in Figure 4.

In Figure 4, the core depressive symptoms index levels are almost parallel for females and males who are carriers of allele T:T, and increase with the level of maltreatment. In males carrying C:C or C:T, the slope of the line has a steeper incline with increasing maltreatment, while for females carrying C:C or C:T, the core depressive symptoms index level decreases with increasing maltreatment.

#### 3.3.3. Moderation Model 3 with Outcome Core Depressive Symptoms Index in 2018

A moderation model 3 analogous to the one described in Figure 2, with the outcome variable Y = core depressive symptoms reported in 2018, was completed for the three-way-interaction. However, we found n = 1061, *R*^2^ = 0.253, *p* ≤ 0.001, and the addition of the interaction did not make a significant change (*R*^2^ change = 0.003, *p* = 0.06); with the Johnson–Neyman method, no transition points were observed in the range of family maltreatment.

Linear regression analyses with outcome of sleep complaints index in 2018, and logistic regression with outcome of thoughts of death and suicide in 2018, did not show statistical significance of the interaction effect. Results can be seen in Appendix A, Table A5 and Table A6.

### 3.4. Analysis of Variance Results

#### 3.4.1. Cross-Sectional Analyses

Cross-sectional analyses are presented in Appendix A. The effect of the studied interaction was statistically significant in relation to core depressive symptoms in 2012. Similarly, as the result with the first, method generalized linear regression. Table A7 shows that, consistent with the above reported regressions, the relationship between the three-way interaction and the sleep complaints index outcome was not statistically significant. Table A8 shows that in both regressions, the mains effects of sex and family maltreatment were significant.

#### 3.4.2. Longitudinal Analyses

Longitudinal analyses are presented in Appendix A, Table A9, Table A10, Table A11 and Table A12. In Table A9, the relation between the three-way interaction and the core depressive symptoms in 2015 was not statistically significant, even though sex, family maltreatment, and the interaction of sex and family maltreatment were significant. Table A11 shows that the relation between the three-way interaction and core depressive symptoms in 2018 had a similar trend to that obtained with the previous method, and a significant main effect of sex.

The results of complementary analyses to control the significance of the interaction substituting PER2 *rs56013859* with AUTS2 *rs69343555* were not statistically significant.

## 4. Discussion

The purpose of this study was to analyze the predicted three-way interaction of sex, *rs56013859* polymorphism, and family maltreatment in relation to depressive symptoms in adolescents. A three-way interaction was found. According to the proposed models, the present study found effects of the three-way interaction in both the cross-sectional and longitudinal analyses in relation to core depressive symptoms in participants aged 13–15 years with both methods, and six years later.

There is previous research on *rs56013859* in relation to alcohol consumption [57], and sleep problems. The study of *rs56013859* polymorphism in relation to family maltreatment, depressive symptoms, sleep complaints, and thoughts of death and suicide is novel, and we are not aware of previous studies that have investigated this relationship.

There is a great deal of literature about the relationship between depressive symptoms and adverse experiences in the family environment [52,94]. Family maltreatment itself plays an important role in the development of depressive symptoms [95]. Some studies have reported mixed results about the effect of genes that regulate the circadian cycle on affective symptoms [54,67,96,97].

Females aged 13–15 years, who were carriers of the *rs56013859* C:C/C:T polymorphism and who were exposed to family maltreatment, presented higher levels of core depressive symptoms than females with *rs56013859* T:T. On the other hand, among male carriers of C:C/C:T, a lower increase of symptoms in the interaction was found. A complementary finding was the level at which family maltreatment became significant (1.1) in the moderation model 3 at the first time point, although overall the level for maltreatment reported by the studied population was low [86].

Notably, six years later, when the participants were 19–21 years old, regression analyses of the same model of interaction in relation to the core depressive symptoms index found male carriers of C:C/C:T presented higher levels of core depressive symptoms. Although the model used for the second method was not statistically significant, analysis of variance was used.

It is not clear why a reverse trend in the interaction in relation to core depressive symptoms was found at the third time point. We observed a reduction in the number of participants and an increase in the proportion of remaining participants with depressive symptoms, especially among males. A possible reason for the results obtained in 2018 could be changes over time of the intersection of family maltreatment and the presence of depressive symptoms. Additionally, when looking at the results of the descriptive statistics, the mean values of the core depressive symptoms index increased for females and males between 2012 and 2015. However, between 2015 and 2018, this value increased only for males, while for females it remained around the same value. Additionally, in 2018, the proportion of females who were carriers of the minor allele and showed depressive symptoms decreased in comparation to males. This may be due first to an effect of the population distribution, and second, to a relatively decreased importance of the family maltreatment in comparison to other risk factors added over time. Furthermore, we noticed a comparative decrease in the family maltreatment index. The changes in both the independent and the dependent variables modified the effect of the interaction in the course of time. Additionally, the sum of more positive or negative environmental factors can modify the level of the depressive symptoms [98]. As well as the development of coping strategies in individuals [99], changes in the transcription of the gene or a possible interaction with other genes (G × G × E) [100], such as with MAOA [56], may help explain the trend. Indeed, it has been suggested that age and sex in combination with genotype have an influence on MAOA gene methylation [101]. Additionally of interest is the influence of Per2 on the expression of MAOA and changes in the regulation of the mesolimbic dopaminergic system in an animal model [56].

With regards to the effect of sex and gene polymorphisms, a sex-specific effect of the interaction of the MAOA gene promoter (VNTR) polymorphism and self-reported maltreatment in relation to adolescent delinquent behavior has been found [82]. A similar sex and age difference was reported by Comasco et al. [58] for the *rs56013859* polymorphism, showing diverse effects in relation to alcohol consumption and sleep problems described, although they used different age/populations rather than a cohort study. Some studies have also described differences between the sexes in the expression of candidate genes for depression, but the role of sex in heritability is not yet clear [72,102,103].

Another aspect to take into account during adolescence is the change in levels of estrogens and testosterone [104] and the presence of the estrogen receptor β which, with the corticotropin-releasing-factor-bearing cells in the paraventricular nucleus of the hypothalamus, control the response to stress [105]. Additionally, stress generates sex differences in the transcriptional profile in all brain regions in humans with MDD and in animals [106]. The monoamine neurotransmitters, dopamine, serotonin, and norepinephrine, have been related to the presence of depressive symptoms [107], and the hypothesis of the relationship between levels of monoamines and depressive symptoms has been behind the use of antidepressant treatments [108]. Moreover, exposure to stress has also been associated with changes in the metabolism of amines [48]. These findings suggest a stress-diathesis explanation of depression [109].

Additionally, epigenetic changes are induced by stressful and traumatic events in the modification of the transcriptional, translational, and posttranslational regulation [110], in conjunction with the fact that *rs56013859* is located in an intron [61,111], and there are several candidate genes that converge through epigenetic mechanisms [112].

Altogether, these findings suggest that if there is a major susceptibility for the risk variant allele in females exposed to a stressful environment, then the variability in the codification of proteins by this intron could be related to levels of glutamate [57,69,72,113].

Spanagel et al. [57] hypothesized that the increased alcohol consumption of the Per2^Brdm1^ mutation mice is related to a reduction in the glutamate excitatory amino acid transporter 1 (Eaat 1), in theory reducing the clearance of glutamate in the synaptic cleft, which combined with a compensatory upregulation of Eaat 2, results in elevated glutamate levels in mutant mice [114]. In addition, Blomeyer et al. [72] proposed that stressful experiences lead to negative feelings, and drinking is used to cope with stress, and that later the consumption of alcohol alters the response to stress and disturbs the circadian rhythmicity. They examined the role of the *rs56013859* genotype in moderating stressful life experiences by drinking. Although they suggested a protective effect of the minor allele against alcohol use, no evidence for the exact mechanism by which it would work like a protector was presented. A glutamatergic hypothesis was also mentioned through regulation in a transcription factor of PER2. An important point to highlight is that the present study examines at interactions in relation to depressive symptoms and not to alcohol use.

Moreover, clinical and preclinical evidence exists regarding the relation of the glutamatergic system with the physiopathology of depression [114,115,116,117]. Hyperglutamatergic neurotransmission has been associated with depression [118]. Dysregulation of the glutamatergic system could be related with depressive symptoms by excess or by defect.

There is also an association between glutamate levels and chronic stress. In mice, chronic stress reduces the glutamatergic signal in the frontotemporal cortex generating symptoms similar to depression [119]. Adding up the results of the literature with the results obtained in our study, we propose the glutamatergic hypothesis as an implied pathway.

In connection with the effect of the interaction on depressive symptoms and changes over time, the depression scale allowed a minimum period of two weeks to be registered. However, depressive symptoms of a chronic nature would be more likely reported in the measurements. An additional factor is the possible change of the moderation effect of the environment with increasing age [120].

When we looked at separate groups of depressive symptoms, we were unable to find a statistically significant relationship between the three-way interaction and sleep complaints, despite the apparently good reliability of the scale used, and the statistically significant differences in sleep complaints and core depressive symptoms between the sexes in the sample. This was contrary to our expectations based on studies linking sleep disorders and *rs56013859* [58]. Family maltreatment seemed to be related to sleep complaints in both longitudinal analyses in accordance with Schønning et al.’s [121] findings.

It was also not possible to find a statistically significant relationship between the three-way interaction and thoughts of death and suicide. Reasons for that could be the low relative frequency of thoughts of death and suicide in this community sample. However, it is important to highlight the relation between family maltreatment and thoughts of death and suicide at the time of both measures, which is consistent with findings of previous studies [122].

In brief, despite hypotheses suggesting a possible connection of *rs56013859* with sleep complaints and the alleged effects of sleep on depression and suicidal ideation, the present study did not confirm such an intricate pattern. In the present study, we used separate models to investigate the proposed three-way interaction of sex, *rs56013859*, and maltreatment in relation to the hypothetical outcome variables. We found the G × E model to be relevant only to the relationship with core depressive symptoms, although with a small effect size as expected by the translation of results in similar interaction studies [91].

### Strengths and Limitations

The study was performed in a community sample cohort, which allowed the realization of measures through time and the possible identification of depressive symptoms that could have a long course. On the other hand, the progressive reduction in the number of participants and the power of the sample was a concern because of the risk of possible errors in interpretation. However, even though there was a decline in participants, there are very few longitudinal studies investigating gene–environment interactions.

The scales used were measured and showed acceptable psychometric properties.

Although family maltreatment and depressive symptoms were not normally distributed in the sample, the size of the sample and the extension of scales were sufficient to permit the use of linear regressions. To control for heteroskedasticity, a robust estimator was used. In the preliminary visual examination of the relation of the variables, a curvilinear relationship was found; nevertheless, the use of an alternative statistical method, or the use of a control including the square of the environment variable in the interaction model would not improve the predictive capacity of the statistical model in relation to depressive symptoms [91].

Depression is a polygenic disorder with low heritability and multiple candidate genes, which requires large study samples that give great importance to environment events [123]. It is therefore difficult to measure the specific effect of this interaction. In addition, the low frequency of the minor allele required the study to be collapsed with the heterozygote group, modifying the possibility of observation of the individual effect [124]. Thus, the effect of the change in the models was small as in many previous gene–environment interaction studies [91].

Moreover, multiple tests could lead to type I errors. However, the study was explorative and the use of a correction (i.e., Bonferroni) was not recommended [89]. We found the predicted relationship between the independent variables and the primary outcome variable, while no such relationship was observed in the two other outcome variables. The model was also robust when using different statistical methods and when re-analyzed in the same population six years later.

Symptom registration was based on the self-report of participants based on limited periods, but the measure scales used for depressive symptoms showed good validity and reliability [75]. The study of diverse maltreatment and violence in the family was more complex, with the scale composed of items concerning violence between parents and violence against the adolescent. However, the scale has been used in several previous G × E studies and has consistently shown to be a solid operationalization of a negative family environment in interactions with various candidate genes [125,126].

A further limitation is that biomarkers, such as hormones (e.g., hypothalamic-pituitary-adrenal axis hyperactivation, cortisol) or neurotransmitters (e.g., monoamines or glutamate) were not measured [127].

The observations in the study are too limited for an understanding of the molecular mechanisms involved. Therefore, it would be necessary to design experimental studies for functional analysis, in order to better understand the pathways through which rs56013859 is associated with depressive symptoms [128].

Identifying the effects of the intron sequences is difficult, since introns can regulate the amount of mRNA for the nucleotides downstream, modifying the function of the gene during transcription [62]. In addition, it is not possible to rule out the effect of a gene with chromosome location in the vicinity of *rs56013859* [129].

The absence of previous studies of *rs56013859* in relation to depressive symptoms makes it necessary to be cautious with these results, and advisable to do further studies.

## Figures and Tables

**Figure 1 genes-14-01723-f001:**
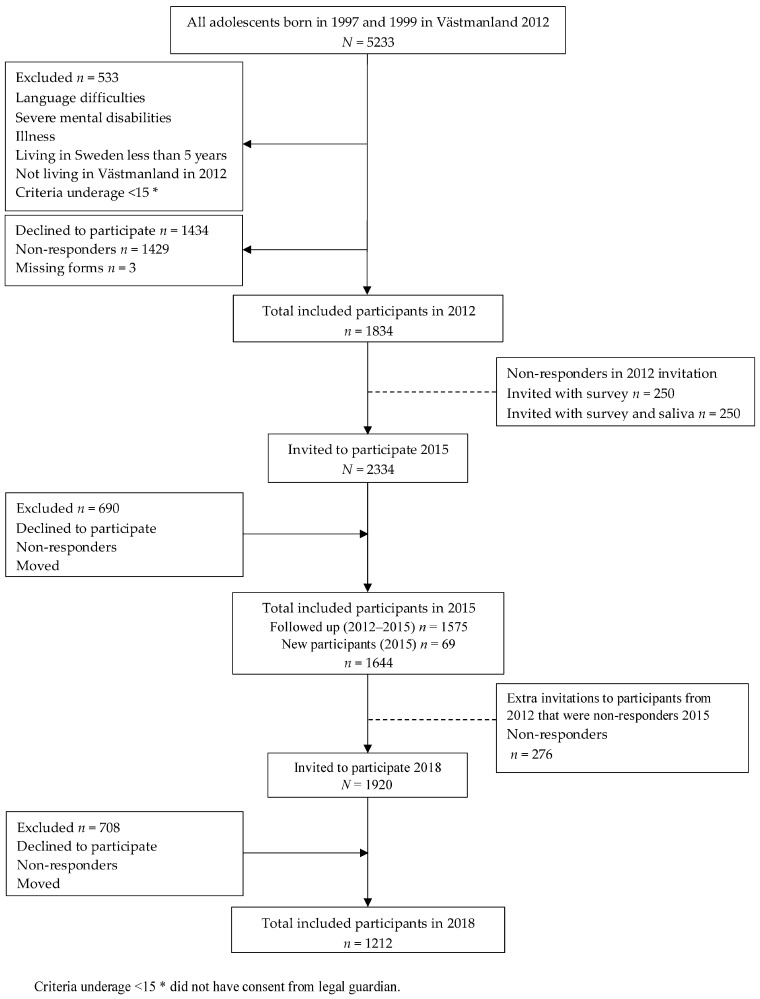
Process for obtaining participants from the SALVe Cohort.

**Figure 2 genes-14-01723-f002:**
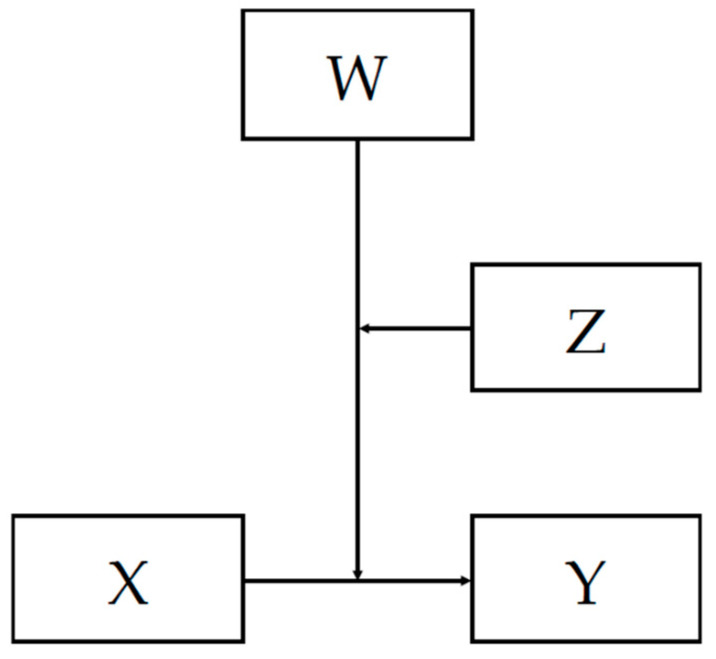
Moderation model 3 where X = *rs56013859* (reference T:T), Y = core depressive symptoms, W = sex (reference male), and Z = family maltreatment in 2012.

**Figure 3 genes-14-01723-f003:**
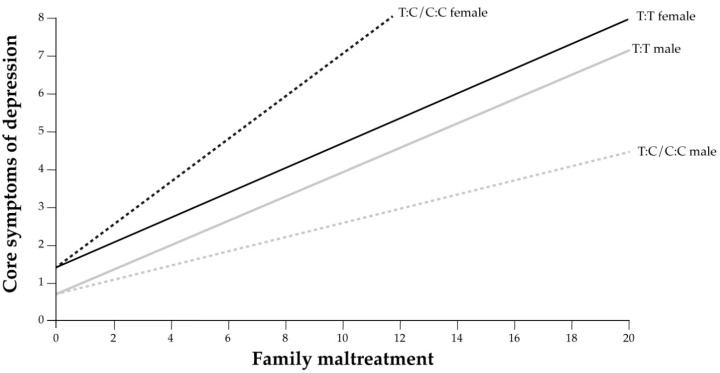
The regression of the interaction of sex and *rs56013859* in relation to the family maltreatment index (X) and to the core depressive symptoms index (Y) in 2012.

**Figure 4 genes-14-01723-f004:**
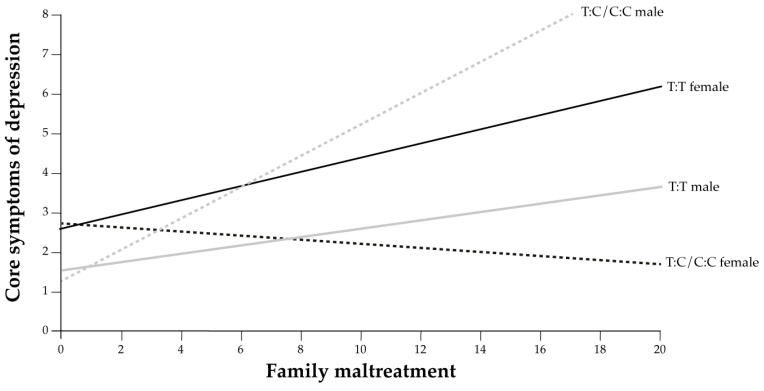
The regression of the interaction of sex and *rs56013859* in relation to the family maltreatment index (X) and to the core depressive symptoms index (Y) in 2018.

**Table 1 genes-14-01723-t001:** Descriptive statistics of the study samples in 2012, 2015, and 2018.

**2012 sample** (*n* = 1834)					
	Missing *n* (%)	*n* (%)	Female	Male	X^2^/Z (*p*)
Sex ^b^ (%)			1019 (56)	815 (44)	
Age, years ^a^ (SD)			14.4 (1)	14.4 (1)	*NS*
Thoughts of death and suicide ^b^	7 (0.4)				
No			893	750	
Yes		184 (10.1)	124 (6.8)	60 (3.2)	***
Core depressive symptoms index ^a^ Mean (SD)	22 (1.2)		1007 1.737 (2.184)	805 0.906 (1.495)	***
Sleep complaints index ^a^ Mean (SD)	151 (8.23)		928 12.831 (9.435)	755 10.535 (8.065)	***
Family maltreatment index ^a^ Mean (SD)	33 (1.86)		964 4.875 (1.905)	775 4.698 (1.375)	*NS*
PER2: *rs56013859* ^b^	157 (8.6)				*NS*
T:T (%)		1288 (76.8)	733 (77.5)	555 (75.9)	
C:T/C:C (%)		389 (23.2)	213 (22.5)	176 (24.1)	
**2015 sample** (*n* = 1644)					
	Missing *n* (%)	*n* (%)	Female	Male	X^2^/Z (*p*)
Sex ^b^ (%)			961 (58.4)	683 (41.6)	
Age, years ^a^ (SD)	2 (0.12)		17.3 (1)	17.3 (1)	*NS*
Core depressive symptoms index ^a^ Mean (SD)	24 (1.5)		943 2.787 (2.49)	667 1.3 (1.86)	***
Sleep complaints index ^a^ Mean (SD)	64 (3.9)		928 20.125 (10.689)	615 15.081 (9.958)	***
Family maltreatment index ^a^ Mean (SD)	87 (5.29)		901 0.842 (1.843)	656 0.675 (1.358)	*NS*
PER2: *rs56013859* ^b^	184 (11.2)				*NS*
T:T (%)		1122 (76)	961 (77)	461 (76.6)	
C:T/C:C (%)		338 (23)	184 (23)	141 (23.4)	
**2018 sample** (*n* = 1212)					
	Missing *n* (%)	*n* (%)	Female	Male	X^2^/Z (*p*)
Sex ^b^ (%)			753 (62.1)	459 (37.9)	
Age years ^a^ (SD)			20.4 (1)	20.4 (1)	*NS*
Thoughts of death and suicide ^b^					
No			605	378	
Yes		201 (19.25)	148 (12.2)	81 (6.7)	*NS*
Core depressive symptoms index ^a^ Mean (SD)	1 (0.01)		752 2.778 (2.584)	459 1.593 (2.275)	***
Sleep complaints index ^a^ Mean (SD)	8 (0.7)		749 20.427 (10.972)	455 17.317 (10.068)	***
Family maltreatment index ^a^ Mean (SD)	53 (4.37)		715 0.857 (1.91)	444 0.664 (1.328)	*NS*
PER2: *rs56013859* ^b^	123 (10.15)				*NS*
T:T (%)		828 (76)	520 (76.1)	308 (75.9)	
C:T/C:C (%)		261 (24)	163 (23.9)	98 (24.1)	

^a^ Analyzed as continuous data, mean, SD: standard deviation; ^b^ Analyzed as categorical data, n (%); *NS*: not statistically significant; *** *p* < 0.001.

**Table 2 genes-14-01723-t002:** Linear regression of the relation of the three-way interaction to the core depressive symptoms index in 2012.

Independent Variables	*p*	*β*	(95% CI)
Sex	<0.001	1.905	(1.553 to 2.336)
*rs56013859*	0.682	0.949	(0.74 to 1.218)
Family maltreatment	<0.001	1.379	(1.214 to 1.566)
Sex ^1^ × *rs56013859* ^2^	0.844	1.042	(0.69 to 1.576)
Sex ^1^ × Family maltreatment	0.925	1.008	(0.851 to 1.194)
*rs56013859* ^2^ × Family maltreatment	0.211	0.875	(0.71 to 1.078)
Sex ^1^ × *rs56013859* ^2^ × Family maltreatment	0.025	1.451	(1.048 to 2.009)
χ^2^ = 231.685	<0.001		

*n* = 1634; *β*, standardized regression coefficient; CI, confidence interval; Sex ^1^, males reference; *rs56013859*
^2^, T:T reference.

**Table 3 genes-14-01723-t003:** Linear regression of the relation of the three-way interaction to the core depressive symptoms index in 2018.

Independent Variables	*p*	*β*	(95% CI)
Sex	<0.001	2.845	(1.958 to 4.135)
*rs56013859*	0.292	0.74	(0.423 to 1.294)
Family maltreatment	0.308	1.111	(0.908 to 1.358)
Sex ^1^ × *rs56013859* ^2^	0.258	1.544	(0.727 to 3.281)
Sex ^1^ × Family maltreatment	0.52	1.077	(0.859 to 1.349)
*rs56013859* ^2^ × Family maltreatment	0.172	1.34	(0.881 to 2.038)
Sex ^1^ × *rs56013859* ^2^ × Family maltreatment	0.044	0.592	(0.355 to 0.986)
χ^2^ = 70.428	<0.001		

*n* = 1061; *β*, standardized regression coefficient; CI, confidence interval; Sex ^1^, males reference; *rs56013859*
^2^, T:T reference.

## Data Availability

The data are not publicly available due to confidentiality.

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
