# Peer review of "A Three-Way Interaction of Sex, PER2 rs56013859 Polymorphism, and Family Maltreatment in Depressive Symptoms in Adolescents"

_genes, 2023, doi:10.3390/genes14091723_

Round 1
Reviewer 1 Report
In this study, the authors studied the three-way interaction of sex, family maltreatment and PER2 SNP in a cohort with major T and minor C alleles at rs56013859 in relation to depressive symptoms. They also discovered female C allele carriers with maltreatment were more prone to have depressive symptoms at early age, whereas males showed symptoms at later age. This study provided detailed experiment design, sample description and thorough discussion on alternative explanations and limitations. It provides valuable information on the relation of T/C SNP of PER2 rs56013859 with depressive symptoms. I only have one suggestion on the introduction and discussion part. Given that the SNP is in the intron of PER2, its impact could be through epigenetic regulation on distal genes, noncoding RNA or PER2 itself. The authors could point out the possibilities in introduction and be explicit in discussion, addressing the need for functional studies and validations.
Reviewer 2 Report
The authors performed cross-sectional and longitudinal analyses with the linear/logistic regression analyses to study a specific genetic variant (PER2 rs56013859), sex, and family maltreatment in relation to core depressive symptoms, sleep issues and thoughts of death in adolescents. The longitudinal analyses over six years revealed a significant three-way interaction to core depressive symptoms where females having specific genetic alleles demonstrating higher depressive symptoms when exposed to family maltreatment. The study revealed novel aspects of circadian rhythm gene in neurological disorders and suicide ideation, however, additional efforts are required to improve the analyses and data interpretation for a more solid conclusion.
1. What types of mental disabilities or disorders could be for the participants who were excluded from the cohort? Does the allele C/T overrepresented in these excluded individuals?
2. Why rs56013859 was specifically selected for this study? Given the variant is intronic, how did the authors prioritize it over numerous other promoter variants for analyses?
3. Is any clinical measurement of depressive symptoms for these participants other than the questionnaire-based assessment available? If so, how did it correlate with the genotypes, sex, or exposure to family maltreatment?
4. When performing the regression analyses, additional control experiments such as irrelevant SNPs need to be included to conclude whether the significant there-way interactions could be exclusive to rs56013859.
5. The author mentioned the glutamatergic theory of depression where the Per2 deletion increased the alcohol consumption and resulted in elevated glutamate in mice (ref. 57). However, Blomeyer et al. concluded that the minor allele at rs56013859 (G) were less engaged in alcohol use, suggesting the allelic effects the author claims could have the opposite effects. How did the author reconcile the conflict observations?
6. The results from 2012 and 2018 shows marginal significance on the three-way interaction terms. However, the term is not significant in data from 2015. Though authors have performed variance analyses, further analyses and interpretation are needed to confirm that the findings are not due to fluctuations or random effects. Additionally, the association of depressive symptoms to family maltreatment from data in 2018, which needs to be further illustrated.
7. The author need to discuss further to clarify why the minor allele C has opposite effects on sex between data in 2012 vs. 2018.
The manuscript is overall easy to follow. However, too many information has been included in the introduction part. Each section is separated and the authors may consider connecting each section of the introduction with stronger logical flows.
Round 2
Reviewer 2 Report
So far I have no more questions to this manuscript